# Shared genetic loci between depression and cardiometabolic traits

**Kristin Torgersen**[1]*, **Zillur Rahman**[2], **Shahram Bahrami**[2], **Guy Frederick Lanyon Hindley**[2], **Nadine Parker**[2], **Oleksandr Frei**[2,3], **Alexey Shadrin**[2], **Kevin S. O'Connell**[2], **Martin Tesli**[2,4], **Olav B. Smeland**[2], **John Munkhaugen**[1,5], **Srdjan Djurovic**[6,7], **Toril Dammen**[8,9], **Ole A. Andreassen**[2]*

**1** Department of Behavioral Medicine and Faculty of Medicine, University of Oslo, Norway, **2** NORMENT: Norwegian Centre for Mental Disorders Research, University of Oslo and Oslo University Hospital, Oslo, Norway, **3** Center for Bioinformatics, Department of Informatics, University of Oslo, Oslo, Norway, **4** Department of Mental Disorders, Norwegian Institute of Public Health, Oslo, Norway, **5** Department of Medicine, Drammen Hospital, Drammen, Norway, **6** Department of Medical Genetics, Oslo University Hospital, Oslo, Norway, **7** NORMENT, Department of Clinical Science, University of Bergen, Bergen, Norway, **8** Section of Psychiatric Treatment Research, Division of Mental Health and Addiction, Oslo University Hospital, Oslo, Norway, **9** Institute of Clinical Medicine, University of Oslo, Norway

\* k.s.torgersen@medisin.uio.no (KT); o.a.andreassen@medisin.uio.no (OAA)

**Data Availability Statement:** We have used GWAS summary statistics. The consortia that we based our study on (the original GWASs) have GWAS summary statistics available. The full GWAS

## Abstract

Epidemiological and clinical studies have found associations between depression and cardiovascular disease risk factors, and coronary artery disease patients with depression have worse prognosis. The genetic relationship between depression and these cardiovascular phenotypes is not known. We here investigated overlap at the genome-wide level and in individual loci between depression, coronary artery disease and cardiovascular risk factors. We used the bivariate causal mixture model (MiXeR) to quantify genome-wide polygenic overlap and the conditional/conjunctional false discovery rate (pleioFDR) method to identify shared loci, based on genome-wide association study summary statistics on depression (n = 450,619), coronary artery disease (n = 502,713) and nine cardiovascular risk factors (n = 204,402–776,078). Genetic loci were functionally annotated using FUnctional Mapping and Annotation (FUMA). Of 13.9K variants influencing depression, 9.5K (SD 1.0K) were shared with body-mass index. Of 4.4K variants influencing systolic blood pressure, 2K were shared with depression. ConjFDR identified 79 unique loci associated with depression and coronary artery disease or cardiovascular risk factors. Six genomic loci were associated jointly with depression and coronary artery disease, 69 with blood pressure, 49 with lipids, 9 with type 2 diabetes and 8 with c-reactive protein at conjFDR < 0.05. Loci associated with increased risk for depression were also associated with increased risk of coronary artery disease and higher total cholesterol, low-density lipoprotein and c-reactive protein levels, while there was a mixed pattern of effect direction for the other risk factors. Functional analyses of the shared loci implicated metabolism of alpha-linolenic acid pathway for type 2 diabetes. Our results showed polygenic overlap between depression, coronary artery disease and several cardiovascular risk factors and suggest molecular mechanisms underlying the association between depression and increased cardiovascular disease risk.

summary statistics for the 23andMe discovery data set will be made available through 23andMe to qualified researchers under an agreement with 23andMe that protects the privacy of the 23andMe participants. Please visit https://research.23andme.com/collaborate/#dataset-access/ for more information and to apply to access the data. This study used openly available software and code, specifically LD-score regression [https://github.com/bulik/ldsc/] (if we report genetic correlation), MiXer (https://github.com/precimed/mixer) and conjunctional FDR [https://github.com/precimed/pleiofdr/].

**Funding:** KT received founding from the University of Oslo, Research Council of Norway (223273, 248778, 273291, 229129, 213837,), KG Jebsen Stiftelsen, South East Norway Health Authority (2017-112, 2019-108), European Union's Horizon2020 Research and Innovation Action Grant # 847776 CoMorMent The funders had no role in study design, data collection and analysis, decision to publish, or preparation of the manuscript.

**Competing interests:** I have read the journal's policy and the authors of this manuscript have the following competing interests: OAA received speaker's honorarium from Lundbeck and Sunovion, and is a consultant to HealthLytix.

## Author summary

Studies have found associations between depression and both cardiovascular disease risk factors and worse cardiovascular disease prognosis. It is unknown if shared genetic factors contribute to these associations. We applied novel statistical tools for polygenic architectures to investigate if there are common genes for depression, coronary artery disease and cardiovascular risk factors (body mass index, blood pressure, lipids, type 2 diabetes and c-reactive protein). We used the statistical methods the bivariate causal mixture model (MiXeR) and the conditional/conjunctional false discovery rate (pleioFDR) to quantify genome-wide overlap and to identify shared genetic variants. We found extensive genetic overlap. Depression shared 68% of genetic variants with body mass index and 14% and with systolic blood pressure. We also identified 79 unique genomic variants associated with depression and coronary artery disease or the risk factors. Variants associated with increased risk for depression also increased the risk of coronary artery disease, some of the lipids and c-reactive protein levels, while there was a mixed pattern of direction for the other risk factors. Further analyses identified shared genetic variants found in metabolism of alpha-linolenic pathway for type 2 diabetes.

## Introduction

Depression is a leading cause of disability worldwide (WHO 2020), and the prevalence of depressive disorder is over 264 million people worldwide (WHO 2017). The association between depression and coronary artery disease (CAD) is well- established [1–8]. The prevalence of clinically significant symptoms of depression in patients with acute coronary syndromes is as high as 30–45% whereas 20% meet the Diagnostic and Statistical manual of Mental Disorder diagnostic criteria for major depression [9]. «Depression» in this paper refers to both clinically significant symptoms as well as major depression and a diagnosis of major depressive disorder. Meta-analyses from hospitals or community groups have reported depression rates of up to 45–51% in CAD patients [9]. This is far higher than that of 4.3% in the general population (WHO 2017). Meta-analyses of depression as a risk factor for CAD in individuals without previous CAD at baseline [6,10–15] reported a significant relative risk of 1.3–2.0 [6,10–15]. Other meta-analyses found depression to be associated with a relative risk of 1.5–2.7 for future cardiac events and all-cause mortality in patients with established CAD [6,12,16–19], and rate ratio of 2 for individuals with mood disorders for dying of circulatory system diseases [20]. Thus, the American Heart Association Science Advisory recommend routine screening of depression in all patients with CAD [21] and for coronary prevention [22]. It is therefore crucial to gain new knowledge about the causes of this comorbidity. Genetic discoveries may also provide important insights for drug development and risk prediction.

While twin studies have found bidirectional associations between depression and CAD [23,24], the specific genes linking depression and CAD are not known. Cause of their association remain unclear (1). Hypothesized mechanisms underlying the association between depression and CAD include subclinical inflammation, endothelial dysfunction, platelet activation, hypothalamic-pituitary-adrenal (HPA) axis hyperactivity, autonomic dysfunction and various behavioral factors such as smoking, obesity, lack of exercise, poor treatment compliance due to depressive symptoms, or maybe related to common factors such as e.g. low education [8,9,25].

Several cardiovascular disease (CVD) risk factors are associated with depression. Epidemiological studies showed that both high and low body mass index (BMI) have been associated with depression [26]. Depression increased the risk of hypertension incidence (RR 1.42), while pooled prevalence estimates of depression in hypertensive patients was 26.8% in a review [27,28]. Evidence that triglyceride (TG) is causally associated with depressive symptoms have been reported [29]. Observational studies have emphasized that higher levels of C-reactive protein (CRP), a general marker of inflammation, is associated with higher score of depressive symptoms [30].

The resent decade's revolution in genetics, with GWAS, in which several hundred thousand to more than a million single nucleotide polymorphisms (SNPs) are assayed in thousands of individuals, has now given new opportunities to assess if there is a shared genetic basis for common psychiatric diseases and CAD risk factors and has motivated us to do this study. Given the high number of traits in humans and the relative small number of genes (~20,000), some genes have to affect multiple traits (genetic pleiotropy) (11). Also, given evidence that depression is associated with ~14,000 variants, it is overly simplistic to reduce its genetic relationship with CVD to a single sum score (i.e. genetic correlation). Therefore we wanted to apply mixer, which is able to uncover more complex genetic relationships which account for both same and opposite genetic effects, as we have identified in several other similar trait.

Genetic risk factors influence both CVD risk factors and CAD as well as depression. Heritability ($h^2$) estimates are 37% for depression [31], 40–55% for CAD [32,33], 47–90% for BMI [34], 15–40% for systolic blood pressure (SBP) and 15–30% for diastolic blood pressure (DBP) [35–37]. GWAS have found 178 loci associated with depression [38], 307 loci associated with CAD [39,40], 536 loci for BMI [41], and 505 independent loci associated with blood pressure [42]. It has been hypothesized that the association between depression and CAD is partly caused by genetic pleiotropy [9], but this has not been evaluated beyond genetic correlations and PRS. While twin studies have found bidirectional associations between depression and CAD [23,24], the specific genes linking depression and CAD are not known and the cause of this association remain unclear (1). Mendelian randomization has indicated that the genetic liability to depression is associated with higher CAD and myocardial infarction risks, partly mediated by type 2 diabetes mellitus and smoking [43]. Furthermore, an increase in polygenic score for depression was associated with CAD [44]. However, while mendelian randomization might point to certain causative relationships [45], it does not identify specific genes underlying the relationship. Genetic liability to depression has been associated with higher CAD and myocardial infarction risks, partly mediated by type 2 diabetes mellitus and smoking [43]. While mendelian randomization might point to certain causative relationships [45], it does not identify specific genes underlying the relationship. Increase in polygenic score for depression was associated with CAD [44]. Previous studies have found minimal genetic correlations ($r_g$ 0.12 ($p = 8.47 \times 10^{-6}$) for CAD, $r_g = 0.13$, ($p = 3.43 \times 10^{-11}$) for BMI, $r_g = 0.11$, ($p = .0001$) for T2D, 0.0108 (p = 0.897) for HDL, -0.189 (p 0.818) for TC and 0.176 (p 0.0203) for TG [25,46]. However, Genetic correlation returns a single estimate of genetic overlap which averages the correlation of effect sizes across all SNPs, after controlling for LD. It is therefore possible for two traits to have a genetic correlation close to 0 despite regions with strong negative and positive correlation. In brain-related traits including depression and intelligence mixed genetic effects may be present [47–49].

In contrast, the bivariate causal mixture model (MiXeR) provides an estimate of the total number of shared and unique variants influencing two traits [50]. I.e. given evidence that depression is associated with ~14,000 variants, it is overly simplistic to reduce its genetic relationship with CVD to a single sumscore (i.e. genetic correlation). Therefore we wanted to apply mixer, which is able to uncover more complex genetic relationships which account for

both same and opposite genetic effects, as we have identified in several other similar trait [48,51,52].

The term "causal" in the bivariate mixture model refers to the cause of the statistical association and not necessarily the biological cause of the conditions studied. This has enabled the discovery of extensive genetic overlap between brain-related traits with non-significant genetic correlation and mixture of variants with same and opposite effect directions [48,52–54]. This is relevant since it may imply shared variance beyond genetic correlation [55]. Determining genetic overlap at the individual locus level is also informative to obtain biological insights. However standard cross-GWAS techniques rely on massive multiple testing, limiting statistical power for discovery [55]. In contrast, the conditional/conjunctional false discovery rate method (cond/conjFDR) leverages information from two GWAS to identify shared loci with likely true effects, and thus increase yield from existing GWAS [47,53,54]. We have recently identified 32 genetic loci shared between depression and BMI, with the majority with a positive direction of effect [52].

We here apply MiXeR to investigate the shared polygenic architecture [50] beyond genetic correlation. Further, we investigate if there are genetic loci jointly associated with depression and CAD as well as a series of CVD risk factors using the cond/conj FDR method [47,53,54]. We applied summary statistics from GWAS of depression [56], CAD [40], body mass index (BMI) [57], systolic blood pressure (SBP) [42], diastolic blood pressure (DBP) [42], high-density lipoprotein (HDL) [58], low-density lipoprotein (LDL) [58], triglycerides (TG) [58], total cholesterol (TC) [58], (T2D) [59], c-reactive protein (CRP) [60].

## Materials and methods

### Ethics statement

We applied to the Regional committees for medical and health research ethics south east Norway (REK) regarding our project, application number 2011/1980 D. As the aim of the project was to test a new statistical model for analysis of anonymous genetic data, the project uses data from already published GWAS of different patient groups, written informed consent already exists for all data in the already published GWAS and that in the present study the data is only treated at a group level REK decided further approval was not necessary. All GWAS used in the present study were approved by the local ethics committees and all the participants gave their written informed consent [40,42,56–60]. UK Biobank received ethical approval from the Research Ethics Committee (REC reference 11/NW/0382). Participants from the 23andMe sample provided informed consent and participated in the research online, under a protocol approved by the external AAHRPP-accredited IRB, Ethical & Independent Review Services (E&I Review) [61]. For more details, see the original publications [40,42,56–60].

### Material

In the present study, GWAS summary statistics data were available on 450,619 individuals (121,198 cases with depression), 329,421 controls without depression), from the psychiatric genomics consortium PGC29 cohort, a GWAS mega-analysis of 29 samples and five additional cohorts all with European-ancestry [56]. More information can be found in S1 Text.

We used GWAS summary statistics results for depression (n = 450,619) [56], CAD (n = 502,713) [40], and 9 CVD risk factors; BMI [57], systolic blood pressure (SBP) [42], diastolic blood pressure (DBP) [42], high-density lipoprotein (HDL) [58], low-density lipoprotein (LDL) [58], triglycerides (TG) [58], total cholesterol (TC) [58], (T2D) [59], c-reactive protein (CRP) [60] (n = 204,402–776,078). More information on the inclusion criteria for the different GWAS is given in the original publications [40,42,56–60].

## Statistical analyses

We used MiXeR to quantify polygenic overlap between depression, BMI, SBP and DBP. This method determines the number of trait-unique and shared causal variants between two traits, and the results are presented as a Venn diagram. "Causal" refers to the *cause of the statistical association" and not necessarily the biological cause of the conditions studied.* MiXeR calculates the genetic correlation between the phenotypes and determines the proportion of shared variants with same effect direction on both phenotypes. The overall measure of polygenic overlap on a 0–1 scale, is quantified by the Dice coefficient. Univariate and bivariate estimates and standard errors are calculated by performing 20 iterations using 2 million randomly selected SNPs for each iteration, and then random pruning at an LD threshold of $r^2 = 0.8$ is conducted. Model fit was based on GWAS z-scores. The model fit for CAD and the other CVD risk factors did not fulfill the criteria and could not provide reliable estimates for MiXeR [50].

We generated conditional Q-Q plots to visualize pleiotropic enrichment [62]. The conditional Q-Q plots compare the association with one trait (e.g. depression) within SNPs strata determined by the strength of association with a secondary trait (e.g. CAD). Pleiotropic enrichment exists if the proportion of SNPs associated with a phenotype increases as a function of the strength of the association with a secondary phenotype, and is shown by a successively leftward deflection from the null line on the conditional Q-Q plot. This can be directly interpreted in terms of the true discovery rate (1-FDR) [53,63,64].

To improve the discovery of genetic variants associated with depression, CAD and CVD risk factors we used the condFDR method [54]. This statistical framework is an extension of the standard FDR, and uses genetic association summary statistics from the primary trait of interest together with those of a secondary phenotype (e.g. CAD). The method re-ranks the test-statistics of the primary phenotype based on the strength of the association with a conditional phenotype, e.g. CAD or a CVD risk factor. We thereby increase the power and incorporate useful information from a second trait into the analysis. Replacing the roles of primary and secondary phenotypes gives the inverse condFDR value. P-values were corrected for inflation using a genomic inflation control procedure [53].

The conjFDR method [53], an extension of the condFDR, was also applied to detect which loci showed evidence of association with both depression and the secondary trait. The conjFDR framework is defined by the maximum of the two condFDR values for a specific SNP. It estimates the posterior probability that a SNP is not associated with either trait or both, given that the P values for both phenotypes are equal to, or smaller, than the P-values for each phenotype individually. We used a condFDR level of 0.01 and a conjFDR of 0.05 per pairwise comparison. The aim of condFDR is to identify new loci, and so a low false positive rate is prioritised. In contrast, the conjFDR analysis is used to identify shared mechanisms, and so a higher false positive rate is tolerated in order to increase the number of true positives and provide greater insights into shared mechanisms. We did the cond/conjFDR analyses of all phenotypes except for BMI, because BMI has recently been analysed by us with the cond/conjFDR methods [52].

Manhattan plots were constructed based on the conjFDR value to show the genetic risk loci shared between depression and the secondary phenotypes. Before the conjFDR analyses SNPs in the extended major histocompatibility complex and 8p23.1 region were excluded. For more details, see the original [53] and subsequent publications [47,55,65,66].

## Genomic loci definition and functional annotation

We defined a genomic locus according to the FUMA protocol (http://fuma.ctglab.nl/) [67], an online platform for functional annotation of SNPs and genes, to define the independent

genomic loci. SNPs with condFDR <0.01 and conjFDR < 0.05 and independent from each other at r2< 0.6 were identified as independent significant SNPs. Lead SNPs were selected in approximate linkage equilibrium with each other at r2< 0.1 To identify distinct genomic loci, all physically overlapping lead SNPs were merged (LD blocks < 250 kb apart). The borders of the genomic loci were determined by identifying all SNPs in LD ($r^2 \geqq 0.6$) with one of the independent significant SNPs in the locus and with cond/conjFDR<0.1. The genomic region containing all candidate SNPs was assessed as a single independent genomic locus. The 1000 Genomes Project reference panel [68] was used to calculate LD information. The directional effects of the loci shared between depression and cardiovascular traits were evaluated by comparing the z-scores of their lead SNPs.

To predict the deleteriousness of SNPs on the proteins structure and function we applied *Combined Annotation Dependent Depletion* [69]. Further, we used *RegulomeDB* [70] to predict regulatory functions, and *chromatin states* to predict transcription and regulatory effects of chromatin states at the SNP locus [71,72]. Candidate SNPs were aligned to genes using the following three strategies implemented in FUMA: Positional gene mapping to align SNPs to genes based on physical proximity, expression quantitative trait locus (eQTL) mapping SNPs to the genes whose expression level is influenced by allelic variation at the SNP level and chromatin states using 3D DNA-DNA interactions to link SNPs to genes. Novelty checking was performed using two strategies. We first identified all loci physically overlapping with the following GWAS and secondary analyses [52,73–78]. We subsequently used the NHGRI-EBI catalog [79] to identify loci overlapping with previously reported GWAS associations. A locus was deemed novel if it was not physically overlapping with any of the reported loci and did not possess any candidate SNPs which have been reported in the GWAS catalogue. We used FUMA [80] for gene-set enrichment for the genes nearest the identified shared loci represented by Gene Ontology [81] was based on this gene mapping strategies. The genotype expression (GTEx) resource [82] were applied to place the likely regulatory lead SNPs in potential biologic context.

## Replication of condFDR and conjFDR significant loci in an independent sample

We tested for evidence of consistent genetic effects in an independent major depression sample from FinnGen, following a similar procedure as described in [51]. The probability of replicating individual loci at genome-wide significance is low due to weak genetic effects. Therefore, we first tested for en masse sign concordance of effect direction in primary and replication samples, in line with previous literature [83–85]. There was significant concordance if the lead SNP had concordant effect directions in the primary and replication depression samples. We used a one-sided exact binomial test of significance under the null hypothesis that significant concordance was randomly distributed [51,83–85].

We then adjusted for smoking using mtCOJO analyses and rerun all the same analyses to see how adjusting for smoking would affect the results.

## Results

### Polygenic overlap

*MiXeR*: The results show evidence of substantial polygenic overlap between depression and BMI. Of 13.9K trait-variants influencing depression and 11K trait-variants influencing BMI, 9.5K (SD 1.0K) were shared. The Dice coefficient was 0.76 (SD 0.08) (Figs 1A and S1). Of 4.4K trait-variants influencing SBP and 4.0K influencing DBP, 2.0K (SD 0.4 K) and 1.0K (SD 0.3K)

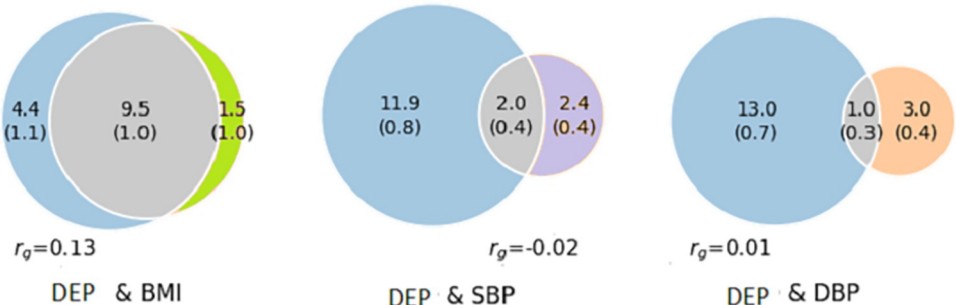

**Fig 1. Venn diagrams of causal shared and unique variants.** The polygenic overlap between a) depression (blue) and BMI (green) b) depression (blue) and SBP (purple) and c) depression (blue) and DBP (orange). The numbers is the quantity of causal variants with standard errors in parentheses (numbers in thousand). DEP; depression, BMI; body mass index, SBP; systolic blood pressure, DBP; diastolic blood pressure, $r_g$; genetic correlation.

were shared with depression, respectively. (Dice coefficient 0.22 (SD 0.04) and 0.11 (SD 0.04) for SBP and DBP, respectively). (Figs 1, S2 and S3). The proportion of variants with concordant effects on SBP and DBP was 24% and 42%, respectively. The model fit for CAD and the other CVD risk factors were not good enough to provide reliable estimates for MiXeR (S1 Text, S4–S10 Figs and S4–S5 Tables).

The MiXeR results of the smoking corrected analyses (mtCOJO) results for depression and BMI were somewhat different from the original MiXeR results. Of 14.3K trait-variants influencing depression and 6.6 K trait-variants influencing BMI, 5.5K (SD 0.9 K) were shared. The Dice coefficient was 0.52 (SD 0.08) (S1 Data). The results of the smoking corrected analyses for depression and SBP and DBP were similar to the results of the original analyses (S1 Data). For more information, see the S1 Text.

Conditional Q-Q plots show polygenic enrichment for DEP|CAD and DEP|all of the risk factors and for CAD|DEP and all of the risk factors|DEP. We observed a pronounced successive leftward deflection for DEP|SBP, DEP|DBP and SBP|DEP and DBP|DEP and for CRP|DEP, TG|DEP, HDL|DEP, LDL|DEP and DEP|T2D. The enrichment pattern in conditional Q-Q plot for DEP|CAD, CAD|DEP, DEP|TC and TC|DEP is less pronounced, but there is clear trend for successive leftward deflection in the "polygenic" or middle region. Also the leftward deflection was less pronounced in DEP|CRP, DEP|LDL, DEP|HDL, DEP|TC, TC|DEP and T2D|DEP. In Q-Q plot we are primarily interested in the behavior of the middle part where the polygenic component resides and occasionally observed erratic trajectories of tails are driven by a handful of variants which do not reflect trend *en masse*. *CondFDR*: After conditioning on each of the secondary traits, we identified 76 loci associated with depression (condFDR < 0.01 and conjFDR <0.05) (S6 Table). Several of these were common for more than one secondary phenotype. CondFDR analysis identified 43 loci associated with depression conditional on CAD, 44 conditional on SBP and 51 conditional on DBP. (S6 Table). The reversed condFDR analysis identified 131, 915 and 924 associated with CAD, SBP and DBP respectively, conditionally on depression. (S7 Table). We also identified depression loci conditional on TC, TG, T2D, LDL, HDL, CRP and vice versa (S6 and S7 Tables).

*ConjFDR*: To identify the genetic loci jointly associated with both depression and CVD risk factors and CAD we used conjFDR. We found 79 unique genetic loci jointly associated with depression and CAD or one or more of the CVD risk factors. Table 1, Figs 2 and S29–S35 and S8–S10 Tables show distinct genomic loci jointly associated with depression and CAD and CVD risk factors, the effect directions and the number of novel loci for depression. In Fig 2 the Manhattan plots for depression and CAD, SBP, LDL are shown as we consider these to be the

**Table 1. Number if distinct genomic loci jointly associated with depression and CAD and CVD risk factors, the effect directions and the number of novel loci for depression.** CAD; coronary artery disease, T2D; type 2 diabetes, CRP; c-reactive protein, HDL; high-density lipoprotein, LDL; low-density lipoprotein, TC; total cholesterol, TG; triglycerides, DBP; diastolic blood pressure, SBP; systolic blood pressure.

| CAD/Cardiovascular associated trait | No of loci | Concordant effect (%) | Novel in depression | No of loci overlapping with condFDR DEP\|CVD (%) | No of loci overlapping with condFDR CVD\|DEP (%) |
|---|---|---|---|---|---|
| CAD | 6 | 83.33 | 1 | 4 (67) | 3 (50) |
| T2D | 9 | 22.22 | 3 | 4 (44) | 3 (33) |
| CRP | 8 | 75.00 | 4 | 3 (37.5) | 4 (50) |
| HDL | 14 | 28.57 | 5 | 6 (43) | 9 (64) |
| LDL | 10 | 55.6 | 1 | 9 (90%) | 5 (50) |
| TC | 11 | 45.45 | 2 | 9 (82) | 8 (73) |
| TG | 14 | 86.00 | 6 | 6 (43) | 13 (93) |
| DBP | 31 | 41.94 | 7 | 16 (52) | 18 (58) |
| SBP | 38 | 24.00 | 12 | 17 (45) | 26 (68) |

most clinically relevant. The Manhattan plots for the other phenotypes can be found in the S1 Text and S29–S35 Figs.

Several loci were common for depression and more than one secondary phenotype (Table 2). Depression and CAD and 5 of the CVD risk factors (T2D, HDL, LDL, TC, TG and DBP) were associated with largely overlapping regions on chromosome 11 with lead SNPs being in strong LD (r2>0.7). Fatty Acid Desaturase 2 (*FADS2) gene* was the nearest gene for all these lead SNPs. Zoom plots centered on these lead SNPs for all the phenotypes can be viewed in S35–S40 Figs.

The distribution of the shared variants can be seen by conjFDR Manhattan plots, where all SNPs without pruning are shown, and the independent significant lead SNPs are encircled in black (Figs 2 and S29–S35).

We then replicated the results of the cond/conjFDR analyses in an independent dataset, FinnGen (https://www.finngen.gitbook.io/documentation/). Applying a significance threshold of 0.05, all the condFDR results were highly significant, and 5 out of 9 conjFDR analyses were significant. All four non-significant conjFDR analyses had a low number of shared loci discovered (<10) which reduced the statistical power of the exact binomial test. Further details and replication results are provided in the S1 Text and S2 Data.

## Effect directions

For the six loci shared between depression and CAD (conjFDR<0.05), five (83%) of the lead SNPs had concordant direction of effect, 12 loci (86%) for TG, 6 loci (60%) for LDL, 6 loci (75%) for CRP which implies that CAD, higher TG, LDL and CRP increase the risk for depression. For depression and SBP, HDL and T2D only 9 (24%), 4 (28.6%) and 2 loci (22%) respectively had concordant effect directions, suggesting that the depression risk was reduced with increased T2D, higher SBP and higher HDL. For TC and DBP there were a mixed pattern of effect direction with same direction of effect in 5 loci (45.5%) and 13 loci (42%), respectively. This is in line with the earlier mentioned genetic correlations, except for T2D ($r_g$ = 0.11, ($p$ = .0001)) and HDL ($r_g$ = 0.0108 (p = 0.897)) which had positive genetic correlations.

Functional annotations of all SNPs having a conjFDR<0.05 for depression and CAD and CVD risk factors are shown in S11 Table. The shared loci revealed variants associated to metabolism of alpha-linolenic acid pathway for T2D (S12 Table). The analyses did not reveal any pathways for the other phenotypes.

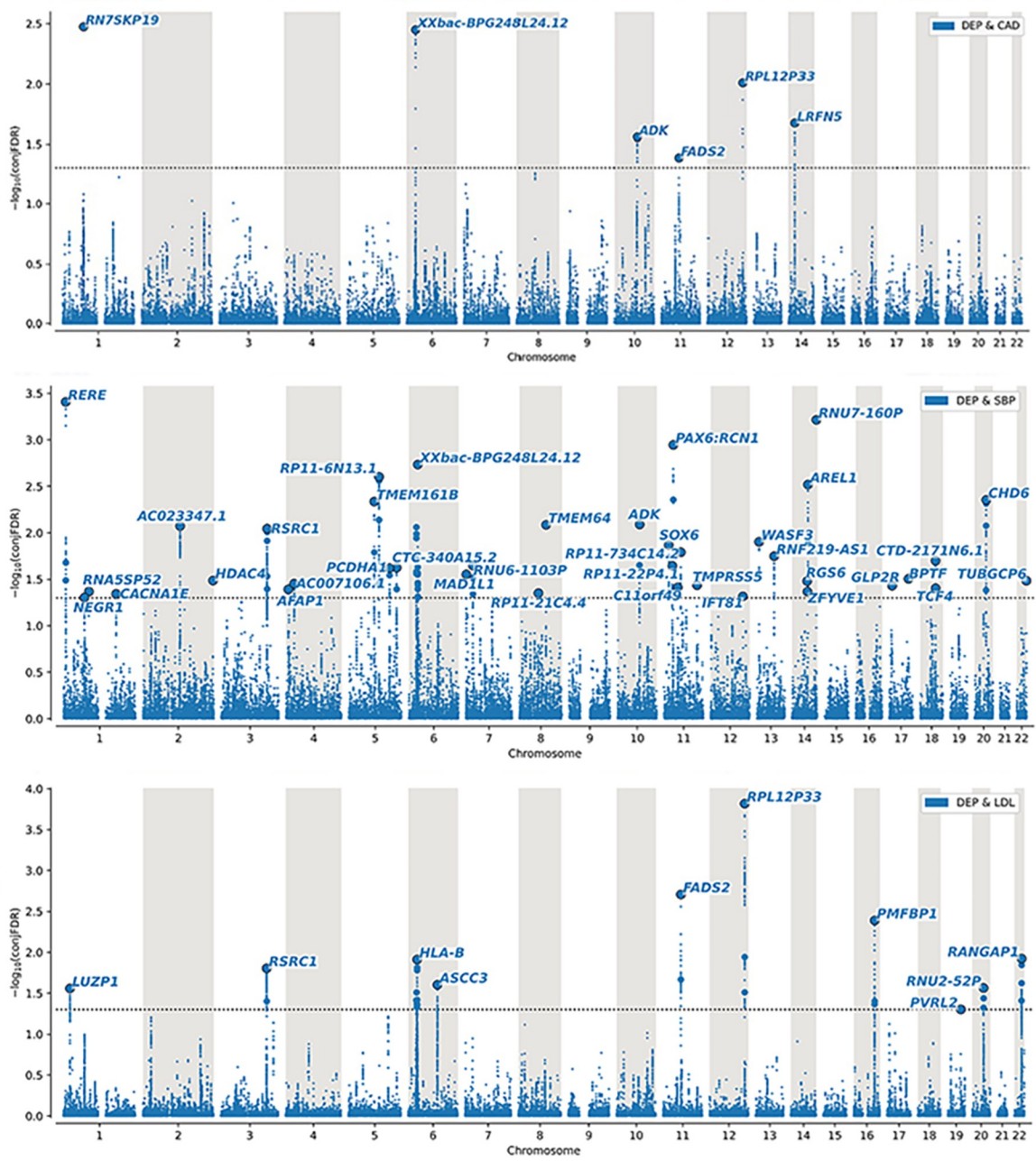

**Fig 2. ConjFDR Manhattan plot.** Common genetic variants both associated with depression (n = 450,619) and a) CAD (n = 502,713), b) SBP (n = 750,000) and c) LDL (n = 297,626) at conjunctional false discovery rate (conjFDR) < 0.05. Manhattan plot showing the–log10 transformed conjFDR values for each SNP on the y axis and the chromosomal positions along the x axis. The dotted horizontal line represents the threshold for significant shared associations (conjFDR < 0.01, i.e. -log10(conjFDR) > 2.0). Independent lead SNPs are encircled in black, and labeled by its nearest gene. The significant shared signal in the major histocompatility complex region (chr6:25119106–33854733) is represented by one independent lead SNP. Further details are available in S8 Table. Dep; depression, CAD; coronary artery disease, SBP; systolic blood pressure, LDL; low-density lipoprotein.

## Discussion

We identified a significant number of genomic loci jointly associated with depression and both CVD risk factors and CAD, implicating overlapping biological pathways. There was a

**Table 2. Loci with common lead SNP for depression and more than one secondary phenotype.** SNP; Single-Nucleotide Polymorphism, CHR; chromosome, MinBP; minimum base position, MaxBP; maximum base position, Dep; depression, CAD; coronary artery disease, T2D; type 2 diabetes, CRP; c-reactive protein, HDL; high-density lipoprotein, LDL; low-density lipoprotein, TC; total cholesterol, TG; triglycerides, DBP; diastolic blood pressure, SBP; systolic blood pressure.

| LEAD SNPs shared for Depression and more than one secondary phenotype at conjFDR<0.05 | | | | | | | |
|---|---|---|---|---|---|---|---|
| CHR | LEAD SNP | MinBP | MaxBP | Nearest Gene | Z score of the SNP in Dep GWAS | Locus novel in Dep | Phenotype |
| 1 | rs301805 | 8404093 | 8895448 | RERE | -5,63 | No | DBP,SBP |
| 2 | rs71441095 | 27961614 | 28284633 | MRPL33:RBKS | -4,05 | Yes | CRP,TC,TG |
| 5 | rs3806843 | 1,4E+08 | 1,4E+08 | PCDHA1:PCDHA2: PCDHA3:PCDHA4: PCDHA5:PCDHA6 | -4,49 | No | DBP,SBP |
| 7 | rs10950398 | 12233848 | 12286050 | TMEM106B | 4,70 | No | HDL,TG |
| 8 | rs62526892 | 91518135 | 91797902 | TMEM64 | -4,87 | Yes | DBP,SBP |
| 11 | rs112181005 | 46708196 | 47367371 | C11orf49 | -4,30 | Yes | CRP,HDL,TG, DBP,SBP |
| 11 | rs174594 | 61539020 | 61624181 | FADS2 | 4,47 | No | HDL,LDL,TC, TG |
| 11 | rs4417256 | 28379460 | 28577867 | RP11-22P4.1 | 4,51 | Yes | DBP,SBP |
| 11 | rs7118275 | 16253187 | 16373664 | SOX6 | 4,69 | Yes | DBP,SBP |
| 12 | rs77741769 | 1,21E+08 | 1,21E+08 | RPL12P33 | 5,57 | No | LDL,TC,CRP |
| 13 | rs536966 | 27087872 | 27146921 | WASF3 | 4,71 | Yes | DBP,SBP |
| 14 | rs10149470 | 1,04E+08 | 1,04E+08 | RNU7-160P | 5,53 | No | DBP,SBP |
| 16 | rs3812984 | 71991135 | 72849510 | PMFBP1 | 4,83 | No | LDL,TC |
| 16 | rs4616299 | 7657373 | 7673819 | RBFOX1 | -4,88 | No | DBP,TG |
| 17 | rs11867618 | 65825248 | 65984537 | BPTF | 4,39 | Yes | TG,SBP,HDL |
| 17 | rs9916772 | 9725091 | 9725888 | GLP2R | 4,32 | Yes | DBP,SBP |
| 18 | rs10503002 | 53057188 | 53164693 | TCF4 | 4,29 | No | DBP,SBP |
| 20 | rs76410441 | 39613707 | 40269840 | CHD6 | 5,03 | No | DBP,SBP |
| 22 | rs9611525 | 41215672 | 42216326 | RANGAP1 | 4,96 | No | LDL,TC |

CHR: Chromosome, MinBP: minimum base position, MaxBP: maximum base position. Locus novel in Dep: Locus novel for depression in our study.

Phenotype: Which phenotypes have the same LEAD SNP.

The build used to determine the base pair position was GRCh37/Hg19

large number of trait-influencing variants between depression and BMI and depression and blood pressure, as revealed with the MiXeR analyses. Variants associated with increased risk for depression were also associated with increased risk of CAD, and higher TG, LDL and CRP levels, while there were an opposite or mixed pattern of effect direction for the other traits. These findings provide new knowledge about the molecular genetic mechanisms shared between depression, CAD and CVD risk.

MiXeR analyses found that depression and BMI shared a substantial number of variants. Of 13.9K trait-variants influencing depression and 11K trait-variants influencing BMI, 9.5K (SD 1.0K) were shared. The proportion of shared genetic variants is greater than what would be expected based on the moderate genetic correlation between these traits ($r_g = 0.13$ for BMI). This may indicate shared genetic mechanisms with mixed effect directions. One possible explanation for this is that BMI is strongly influenced by behaviour, and could thus to some degree be considered a "behavioural/brain-related trait". This may explain the large difference in polygenicity when compared to SBP and DBP which could be considered "somatic traits",

and, by extension, the substantially larger number of shared variants (9.5k vs 1/2K). It is also concordant with previous literature that identified odds ratios in the range of 1.0 to 2.0 for the association between obesity and depression. [86]. The difference in the MiXeR results when we corrected for smoking using mtCOJO analyses may further support this. However Mendelian randomization analyses have provided evidence for a 1.12-fold increase in depression per s.d. of BMI which supports that BMI has a causal effect on depression. Here, we found a substantial shared genetic component between depression and BMI.

Of 13.9K trait-variants influencing depression and 4.4K trait-variants influencing SBP, 2.0K (SD0.4K) were shared. This equals almost 50% of the genomic variants influencing SBP. Of 14K trait-variants influencing depression and 4.0K variants influencing DBP, 1.0K (SD 0.3K) were shared. A recent review and meta-analysis revealed common genes for depression and blood pressure, CAD, HDL, LDL and TC [87]. We identified twice as many overlapping variants with depression and SBP as with depression and DBP. DBP's overlap is about the same as we see for height (1.1 K (SD 0.5 K)), which we consider to be a "negative control" i.e. representing non-specific genetic overlap [48]. The difference in overlap is interesting and in line with an epidemiological study finding depression to be inversely related to SBP, while for DBP the findings were weaker or non-significant [88].

Unfortunately, the model fit for MiXeR for the other phenotypes were not good enough. This may be due to that all these traits (lipids, CAD, T2D and CRP) are very low-polygenic (univariate polygenicities are shown in S2 Table) yet highly heritable, making sampling procedure, which is used in bivariate optimization very "nosy". Technically we could try to suppress this "noise" by increasing the number of samples (default is 20K), but computation burden will grow proportionally to the increment of samples potentially making actual calculation not feasible. Therefore, specific details of current implementation (e.g. random sampling procedure to find optimal parameters) limit application domain of MiXeR model to polygenic traits with reasonable heritability.

The second aspect, suggesting why straightforward increase of samples in optimization is not a good solution for this kind of low-polygenic (oligogenic) traits is that one will expect some peculiar features of genetic architecture (e.g. few notable strong signals and fairly weak "polygenic tail" or strong tendency of strong signals to be co-localized in the genome). These peculiarities may strongly violate assumptions of MiXeR model (which assumes that statistically "causal" variants are uniformly distributed in the genome and all effect sizes are drawn from the same normal distribution) making even seemingly stable estimates senseless.

In this study we identified a portion of the specific genetic loci underlying the shared heritability suggested by LDSR and PRS using the conjFDR method, and increased the number of loci due to the increased power of conditioning on secondary phenotypes.

Depression in our study is genetically associated with lower levels of HDL, as could be expected because higher HDL is associated with lower CAD risk (55). A recent study reports consistent evidence that TG is causally associated with depressive symptoms using Mendelian Randomisation. TC and LDL did not show robust causal relationships with depression [29]. For HDL moderate evidence for causal associations with depression were observed [29]. The same is the case for Mediation analyses showing that CRP mediated the relationship between a PGS for depressive symptoms and somatic symptoms [89]. In our study, lead SNPs in loci associated with depression and CRP had concordant direction of effects. Our finding of extensive shared variants between depression and lipids and depression and CRP suggests there may be a large numbers of variants which impact both phenotypes via pleiotropic mechanisms. This is in direct contravention of one of the key assumptions in Mendelian Randomisation, which may raise questions about its application.

The effect direction was mostly negative for depression and SBP, DBP and T2D, which implicates lower depression risk with higher SBP, DBP and T2D. This is in line with studies finding an inverse association between depression and blood pressure [88,90].

Regarding depression and T2D, previous genetic studies have conflicting results, but have not evaluated effect direction as we have here. Ji et al. have identified a total of 496 shared SNPs associated with both T2D and depression at p-value ≤ 1.0E-07. Functional enrichment analysis showed that the enriched pathways pertained to immune responses (Fc gamma R-mediated phagocytosis, T cell and B cell receptors signaling), cell signaling (MAPK, Wnt signaling), lipid metabolism, and cancer associated pathways [91]. A cross-sectional study using genetic score and LDSR do not support an association between T2D and depression at the clinical and genetic level in a multiethnic population at risk for T2D [92]. Hayes et al (2018) found no significant genetic correlation with T2D, but they identified pleiotropic genetic variations for depressive symptoms and T2D (in the *IGF2BP2*, *CDKAL1*, *CDKN2B-AS*, and *PLE-KHA1* genes [93].

In the original depression GWAS 44 loci were identified [56]. By combining the original depression GWAS (n = 450,619) with the CAD (n = 502,713) and CVD risk factors (n = 204 402–750 000) GWAS, we identified 76 unique loci associated with depression. 33–93% of conjFDR-discovered loci were identified by condFDR. Since conjFDR is the maximum of the two condFDR values and the significance threshold was 0.05 for conjFDR and 0.01 for condFDR, this proportion gives an indication of the strength of association between the set of conjFDR lead SNPs and each phenotype. The current results of genetic pleiotropy for depression and CAD risk may help to reveal mechanistic insights into the shared genetic architecture. Our results are in line with other GWAS findings reporting significant positive genetic correlations between depression and CAD (0.12) and CVD risk factors (BMI, T2D) (0.0031–0.13) [25,46] and a detailed review and analysis of CVD risk genes also associated with mood disorders [87]. Our findings are in opposition to negative significant genetic correlations between severe melancholic depression and CAD (-0.0289) and cardiometabolic traits (-0.00164–0.142) [46], and PRS representing severe melancholic depression was associated with reduced cardiometabolic risk. Our results are also contrary to findings from a recent study that a genetic risk score strongly associated with CAD risk was not associated with depression. However they found that family history of heart disease was associated with a 20% increase in depression risk [94].

The concordant effect direction we found in the present study between depression and CAD, TG, LDL and CRP are in line with the earlier mentioned metabolic and inflammatory mechanisms, endothelial dysfunction and unhealthy behavioral factors contributing in the pathophysiology of both depression and CAD. Our finding of a locus with the same lead SNP for depression and CAD, T2D, HDL, LDL, TC, TG and DBP further supports this. The nearest gene- the Fatty Acid Desaturase 2 (*FADS2) gene* can be found in many tissues. The protein encoded by this gene regulates unsaturation of fatty acids. We also found genes implicating the metabolism of alpha-linolenic acid pathway for T2D. This is line with a recent study reporting that plasma linolenic acid was inversely related to T2D risk [95].

These are important findings as an emerging field of research known as nutritional neuroscience has been focusing on the relationship between diet and depression [96]. The brain is rich in lipids, and dietary fatty acids act within specific brain regions to regulate processes impacting emotional behavior [97]. In depressed patients, daily consumption of omega–3 fatty acid has been shown to stimulate mood elevation [98]. Saturated fats have been suggested to be associated with symptoms of depression in humans [99]. However, we must emphasize that gene-set analyses are still quite unreliable and more research is needed to confirm which mechanisms are underlying the development of depression and CAD.

The mixed direction of effect for several phenotypes we have found here may suggest complex mechanisms not fully elaborated as our pathway analyses suggests, and indicate non-identified subgroups of patients particularly vulnerable to developing both depression and CAD. There may be subgroups of patients within depression, explaining the mixed effect direction in several of the phenotypes in our results. It is well known that depression is associated with both high and low BMI [26]. These patients may represent different subgroups of depressed patients, with various associations to metabolic phenotypes such as BMI, lipids and T2D. Findings from a recent study suggests that the associations with CAD and CVD risk factors varied by depression subtypes [46]. Further research is needed to reveal if this is the case, and may ultimately lay the foundation for more personalized treatment. Our finding of specific loci underlying the comorbidity of depression and CAD may motivate researchers to investigate the molecular mechanisms driving the statistical association identified. Furthermore,a more detailed understanding of the shared genetic architecture may enable improved prediction or identification of subgroups at high risk for both disorders in clinical practice.

Strengths of the present study includes a large number of individuals in each sample, and the use of statistical tools that makes use of several samples to further increase statistical power. Limitations includes that phenotypes are assessed by self-report on only one occasion. Cases were either MDD, MD or self-reported cases of depression, and these may represent different subgroups of patients with different associations with CAD and CVD risk factors, possibly explaining a mixed direction of effects.

In conclusions, the present study identified shared genetic architecture, beyond genetic correlation, between depression and CAD and CVD risk factors, most strongly with BMI and blood pressure. We also revealed 79 specific genetic loci underlying the shared molecular genetic architecture of these conditions suggesting that genetic factors cause some of the comorbidity.

## Supporting information

**S1 Fig. Mixer results for depression (DEP) and body mass index (BMI).**
(TIF)

**S2 Fig. Mixer results for depression (DEP) and systolic blood pressure (SBP).**
(TIF)

**S3 Fig. Mixer results for depression (DEP) and diastolic blood pressure (DBP).**
(TIF)

**S4 Fig. Mixer results for depression (DEP) and coronary artery disease (CAD).**
(TIF)

**S5 Fig. Mixer results for depression (DEP) and c-reactive protein (CRP).**
(TIF)

**S6 Fig. Mixer results for depression (DEP) and high-density lipoprotein (HDL).**
(TIF)

**S7 Fig. Mixer results for depression (DEP) and low-density lipoprotein (LDL).**
(TIF)

**S8 Fig. Mixer results for depression (DEP) type 2 diabetes (T2D).**
(TIF)

**S9 Fig. Mixer results for depression (DEP) and total cholesterol (TC).**
(TIF)

**S10 Fig. Mixer results for depression (DEP) and triglycerides (TG).**
(TIF)

**S11 Fig. Polygenic overlap between coronary artery disease (CAD) and depression (DEP) (CAD|DEP).** Conditional q-q plot of nominal versus empirical–log 10p values (corrected for inflation) in primary trait (CAD) below the standard GWAS threshold of $p < 5 \times 10{-8}$ as a function of significance of association with secondary trait at the level of $p< = 0.1$ $p < = 0.01$ and $p < = 0.001$, respectively. The blue line indicates all SNPs. The dashed line indicate the null hypothesis.
(TIF)

**S12 Fig. Polygenic overlap between depression (DEP) and coronary artery disease (CAD) (DEP|CAD).** Conditional q-q plot of nominal versus empirical–log 10p values (corrected for inflation) in primary trait (DEP) below the standard GWAS threshold of $p < 5 \times 10{-8}$ as a function of significance of association with secondary trait at the level of $p< = 0.1$ $p < = 0.01$ and $p < = 0.001$, respectively. The blue line indicates all SNPs. The dashed line indicate the null hypothesis.
(TIF)

**S13 Fig. Polygenic overlap between systolic blood pressure (SBP) and depression (DEP) (SBP|DEP).** Conditional q-q plot of nominal versus empirical–log 10p values (corrected for inflation) in primary trait (SBP) below the standard GWAS threshold of $p < 5 \times 10{-8}$ as a function of significance of association with secondary trait at the level of $p< = 0.1$ $p < = 0.01$ and $p < = 0.001$, respectively. The blue line indicates all SNPs. The dashed line indicate the null hypothesis.
(TIF)

**S14 Fig. Polygenic overlap between depression (DEP) and systolic blood pressure (DEP) (DEP|SBP).** Conditional q-q plot of nominal versus empirical–log 10p values (corrected for inflation) in primary trait (DEP) below the standard GWAS threshold of $p < 5 \times 10{-8}$ as a function of significance of association with secondary trait at the level of $p< = 0.1$ $p < = 0.01$ and $p < = 0.001$, respectively. The blue line indicates all SNPs. The dashed line indicate the null hypothesis.
(TIF)

**S15 Fig. Polygenic overlap between low-density lipoprotein and depression (LDL) (LDL| DEP).** Conditional q-q plot of nominal versus empirical–log 10p values (corrected for inflation) in primary trait (LDL) below the standard GWAS threshold of $p < 5 \times 10{-8}$ as a function of significance of association with secondary trait at the level of $p< = 0.1$ $p < = 0.01$ and $p < = 0.001$, respectively. The blue line indicates all SNPs. The dashed line indicate the null hypothesis.
(TIF)

**S16 Fig. Polygenic overlap between depression (DEP) and low-density lipoprotein (DEP) (DEP|LDL).** Conditional q-q plot of nominal versus empirical–log 10p values (corrected for inflation) in primary trait (DEP) below the standard GWAS threshold of $p < 5 \times 10{-8}$ as a function of significance of association with secondary trait at the level of $p< = 0.1$ $p < = 0.01$ and $p < = 0.001$, respectively. The blue line indicates all SNPs. The dashed line indicate the null hypothesis.
(TIF)

**S17 Fig. Polygenic overlap between high-density lipoprotein (HDL) and depression (DEP) (HDL|DEP).** Conditional q-q plot of nominal versus empirical–log 10p values (corrected for

inflation) in primary trait (HDL) below the standard GWAS threshold of p < 5 x 10–8 as a function of significance of association with secondary trait at the level of p< = 0.1 p < = 0.01 and p < = 0.001, respectively. The blue line indicates all SNPs. The dashed line indicate the null hypothesis.
(TIF)

**S18 Fig. Polygenic overlap between depression (DEP) and high-density lipoprotein (DEP) (DEP|HDL).** Conditional q-q plot of nominal versus empirical–log 10p values (corrected for inflation) in primary trait (DEP) below the standard GWAS threshold of p < 5 x 10–8 as a function of significance of association with secondary trait at the level of p< = 0.1 p < = 0.01 and p < = 0.001, respectively. The blue line indicates all SNPs. The dashed line indicate the null hypothesis.
(TIF)

**S19 Fig. Polygenic overlap between total cholesterol () andTC depression (DEP) (TC|DEP).** Conditional q-q plot of nominal versus empirical–log 10p values (corrected for inflation) in primary trait (TC) below the standard GWAS threshold of p < 5 x 10–8 as a function of significance of association with secondary trait at the level of p< = 0.1 p < = 0.01 and p < = 0.001, respectively. The blue line indicates all SNPs. The dashed line indicate the null hypothesis.
(TIF)

**S20 Fig. Polygenic overlap between depression (DEP) and total cholesterol (DEP)(DEP|TC).** Conditional q-q plot of nominal versus empirical–log 10p values (corrected for inflation) in primary trait (DEP) below the standard GWAS threshold of p < 5 x 10–8 as a function of significance of association with secondary trait at the level of p< = 0.1 p < = 0.01 and p < = 0.001, respectively. The blue line indicates all SNPs. The dashed line indicate the null hypothesis.
(TIF)

**S21 Fig. Polygenic overlap between triglycerides (TG) and depression (DEP) (TG|DEP).** Conditional q-q plot of nominal versus empirical–log 10p values (corrected for inflation) in primary trait (TG) below the standard GWAS threshold of p < 5 x 10–8 as a function of significance of association with secondary trait at the level of p< = 0.1 p < = 0.01 and p < = 0.001, respectively. The blue line indicates all SNPs. The dashed line indicate the null hypothesis.
(TIF)

**S22 Fig. Polygenic overlap between depression (DEP) and triglycerides (DEP)(DEP|TG).** Conditional q-q plot of nominal versus empirical–log 10p values (corrected for inflation) in primary trait (DEP) below the standard GWAS threshold of p < 5 x 10–8 as a function of significance of association with secondary trait at the level of p< = 0.1 p < = 0.01 and p < = 0.001, respectively. The blue line indicates all SNPs. The dashed line indicate the null hypothesis.
(TIF)

**S23 Fig. Polygenic overlap between type 2 diabetes (T2D) and depression (DEP) (T2D|DEP).** Conditional q-q plot of nominal versus empirical–log 10p values (corrected for inflation) in primary trait (DBP) below the standard GWAS threshold of p < 5 x 10–8 as a function of significance of association with secondary trait at the level of p< = 0.1 p < = 0.01 and p < = 0.001, respectively. The blue line indicates all SNPs. The dashed line indicate the null hypothesis.
(TIF)

**S24 Fig. Polygenic overlap between depression (DEP) and type 2 diabetes (DEP)(DEP| T2D).** Conditional q-q plot of nominal versus empirical–log 10p values (corrected for inflation) in primary trait (DEP) below the standard GWAS threshold of $p < 5 \times 10^{-8}$ as a function of significance of association with secondary trait at the level of $p < = 0.1$ $p < = 0.01$ and $p < = 0.001$, respectively. The blue line indicates all SNPs. The dashed line indicate the null hypothesis.
(TIF)

**S25 Fig. Polygenic overlap between c-reactive protein (CRP) and depression (DEP) (CRP| DEP).** Conditional q-q plot of nominal versus empirical–log 10p values (corrected for inflation) in primary trait (CRP) below the standard GWAS threshold of $p < 5 \times 10^{-8}$ as a function of significance of association with secondary trait at the level of $p < = 0.1$ $p < = 0.01$ and $p < = 0.001$, respectively. The blue line indicates all SNPs. The dashed line indicate the null hypothesis.
(TIF)

**S26 Fig. Polygenic overlap between depression (DEP) sc-reactive protein (DEP)(DEP| CRP).** Conditional q-q plot of nominal versus empirical–log 10p values (corrected for inflation) in primary trait (DEP) below the standard GWAS threshold of $p < 5 \times 10^{-8}$ as a function of significance of association with secondary trait at the level of $p < = 0.1$ $p < = 0.01$ and $p < = 0.001$, respectively. The blue line indicates all SNPs. The dashed line indicate the null hypothesis.
(TIF)

**S27 Fig. Polygenic overlap between diastolic blood pressure (DBP) and depression (DEP) (DBP|DEP).** Conditional q-q plot of nominal versus empirical–log 10p values (corrected for inflation) in primary trait (DBP) below the standard GWAS threshold of $p < 5 \times 10^{-8}$ as a function of significance of association with secondary trait at the level of $p < = 0.1$ $p < = 0.01$ and $p < = 0.001$, respectively. The blue line indicates all SNPs. The dashed line indicate the null hypothesis.
(TIF)

**S28 Fig. Polygenic overlap between depression (DEP) and systolic blood pressure (DEP) (DEP|SBP).** Conditional q-q plot of nominal versus empirical–log 10p values (corrected for inflation) in primary trait (DEP) below the standard GWAS threshold of $p < 5 \times 10^{-8}$ as a function of significance of association with secondary trait at the level of $p < = 0.1$ $p < = 0.01$ and $p < = 0.001$, respectively. The blue line indicates all SNPs. The dashed line indicate the null hypothesis.
(TIF)

**S29 Fig. ConjFDR Manhattan plot.** Common genetic variants both associated with depression (DEP) and high-density lipoprotein (HDL) at conjunctional false discovery rate (conjFDR) < 0.05. Manhattan plot showing the–log10 transformed conjFDR values for each SNP on the y axis and the chromosomal positions along the x axis. The dotted horizontal line represents the threshold for significant shared associations (conjFDR < 0.05, ie., -log10 (conjFDR) > 2.0). Independent lead SNPs are encircled in black. The significant shared signal in the major histocompatibility complex region (chr6:25119106–33854733) is represented by one independent lead SNP.
(TIF)

**S30 Fig. ConjFDR Manhattan plot.** Common genetic variants both associated with depression (DEP) and high-density lipoprotein (HDL) at conjunctional false discovery rate

(conjFDR) < 0.05. Manhattan plot showing the–log10 transformed conjFDR values for each SNP on the y axis and the chromosomal positions along the x axis. The dotted horizontal line represents the threshold for significant shared associations (conjFDR < 0.05, ie., -log10 (conjFDR) > 2.0). Independent lead SNPs are encircled in black. The significant shared signal in the major histocompatibility complex region (chr6:25119106–33854733) is represented by one independent lead SNP.
(TIF)

**S31 Fig. ConjFDR Manhattan plot.** Common genetic variants both associated with depression (DEP) and high-density lipoprotein (HDL) at conjunctional false discovery rate (conjFDR) < 0.05. Manhattan plot showing the–log10 transformed conjFDR values for each SNP on the y axis and the chromosomal positions along the x axis. The dotted horizontal line represents the threshold for significant shared associations (conjFDR < 0.05, ie., -log10 (conjFDR) > 2.0). Independent lead SNPs are encircled in black. The significant shared signal in the major histocompatibility complex region (chr6:25119106–33854733) is represented by one independent lead SNP.
(TIF)

**S32 Fig. ConjFDR Manhattan plot.** Common genetic variants both associated with depression (DEP) and high-density lipoprotein (HDL) at conjunctional false discovery rate (conjFDR) < 0.05. Manhattan plot showing the–log10 transformed conjFDR values for each SNP on the y axis and the chromosomal positions along the x axis. The dotted horizontal line represents the threshold for significant shared associations (conjFDR < 0.05, ie., -log10 (conjFDR) > 2.0). Independent lead SNPs are encircled in black. The significant shared signal in the major histocompatibility complex region (chr6:25119106–33854733) is represented by one independent lead SNP.
(TIF)

**S33 Fig. ConjFDR Manhattan plot.** Common genetic variants both associated with depression (DEP) and high-density lipoprotein (HDL) at conjunctional false discovery rate (conjFDR) < 0.05. Manhattan plot showing the–log10 transformed conjFDR values for each SNP on the y axis and the chromosomal positions along the x axis. The dotted horizontal line represents the threshold for significant shared associations (conjFDR < 0.05, ie., -log10 (conjFDR) > 2.0). Independent lead SNPs are encircled in black. The significant shared signal in the major histocompatibility complex region (chr6:25119106–33854733) is represented by one independent lead SNP.
(TIF)

**S34 Fig. ConjFDR Manhattan plot.** Common genetic variants both associated with depression (DEP) and high-density lipoprotein (HDL) at conjunctional false discovery rate (conjFDR) < 0.05. Manhattan plot showing the–log10 transformed conjFDR values for each SNP on the y axis and the chromosomal positions along the x axis. The dotted horizontal line represents the threshold for significant shared associations (conjFDR < 0.05, ie., -log10 (conjFDR) > 2.0). Independent lead SNPs are encircled in black. The significant shared signal in the major histocompatibility complex region (chr6:25119106–33854733) is represented by one independent lead SNP.
(TIF)

**S35 Fig. Zoom plot for DEP and CAD centered on the variant rs174541, close to the Fatty Acid Desaturase 2 (*FADS2*) gene.**
(TIF)

**S36 Fig. Zoom plot for DEP and DBP centered on the variant rs174576, close to the Fatty Acid Desaturase 2 (*FADS2*) gene.**
(TIF)

**S37 Fig. Zoom plot for DEP and HDL centered on the variant rs174594, close to the Fatty Acid Desaturase 2 (*FADS2*) gene.** This leadSNP was in common for depression, HDL, LDL, TC and TG.
(TIF)

**S38 Fig. Zoom plot for DEP and LDL centered on the variant rs174594, close to the Fatty Acid Desaturase 2 (*FADS2*) gene.** This leadSNP was in common for depression, HDL, LDL, TC and TG.
(TIF)

**S39 Fig. Zoom plot for DEP and TC centered on the variant rs174594, close to the Fatty Acid Desaturase 2 (*FADS2*) gene.** This leadSNP was in common for depression, HDL, LDL, TC and TG.
(TIF)

**S40 Fig. Zoom plot for DEP and TG centered on the variant rs174541, close to the Fatty Acid Desaturase 2 (*FADS2*) gene.** This leadSNP was in common for depression, HDL, LDL, TC and TG.
(TIF)

**S1 Table. Description of the five additional cohorts.**
(DOCX)

**S2 Table. Replication of our results in FinnGen MDD sample.** MDD: Major depressive disorder. P-value of binomial test of conjFDR and condFDR.
(XLSX)

**S3 Table. Number of conjFDR loci at 0.05 in original analyses and after correcting for smoking in mtCOJO analyses.**
(XLSX)

**S4 Table. MiXeR results.**
(XLSX)

**S5 Table. MiXeR results.**
(XLSX)

**S6 Table. CondFDR 0.01 DEP vs Trait2 Loci.**
(XLSX)

**S7 Table. CondFDR 0.01 Trait2 vs DEP Loci.**
(XLSX)

**S8 Table. ConjFDR 0.05 DEP vs Trait2 novelty with overlap.**
(XLSX)

**S9 Table. Concordant effect- count DEP Trait2 Lead.**
(TIF)

**S10 Table. Concordant effect- percent DEP Trait2 Lead.**
(TIF)

**S11 Table. ConjFDR 0.05 DEP vs Trait2 Go.**
(XLSX)

**S12 Table. ConjFDR 0.05 DEP vs Trait2 Path.**
(XLSX)

**S1 Text. Supplementary Text.**
(DOCX)

**S1 Data. Results of mtCOJO analyses corrected for smoking.**
(ZIP)

**S2 Data. Results of replication in FinnGen.**
(ZIP)

## Acknowledgments

We want to thank all the research participants taking part in the GWASs included in this study, including the GWAS of depression from PGC and 23andMe, Inc., and the GWASs of CAD and CAD risk factors. In addition, we want to thank the consortia for making their GWAS summary statistics available, the people who provided DNA samples.

The computations were performed on resources provided by UNINETT Sigma 2 –the National Infrastructure for High Performance Computing and Data Storage in Norway.

## Author Contributions

**Conceptualization:** Kristin Torgersen, Guy Frederick Lanyon Hindley, John Munkhaugen, Srdjan Djurovic, Toril Dammen, Ole A. Andreassen.

**Data curation:** Zillur Rahman, Shahram Bahrami, Guy Frederick Lanyon Hindley, Oleksandr Frei, Alexey Shadrin.

**Formal analysis:** Zillur Rahman, Shahram Bahrami, Guy Frederick Lanyon Hindley, Nadine Parker, Alexey Shadrin, Kevin S. O'Connell.

**Investigation:** Kristin Torgersen, Zillur Rahman, Shahram Bahrami, Guy Frederick Lanyon Hindley, Nadine Parker, Alexey Shadrin, Kevin S. O'Connell, Ole A. Andreassen.

**Methodology:** Zillur Rahman, Shahram Bahrami, Guy Frederick Lanyon Hindley, Nadine Parker, Oleksandr Frei, Alexey Shadrin, Kevin S. O'Connell.

**Project administration:** Kristin Torgersen, Srdjan Djurovic, Toril Dammen, Ole A. Andreassen.

**Resources:** Oleksandr Frei, Ole A. Andreassen.

**Software:** Oleksandr Frei.

**Supervision:** Shahram Bahrami, Kevin S. O'Connell, Martin Tesli, Olav B. Smeland, John Munkhaugen, Srdjan Djurovic, Toril Dammen.

**Validation:** Zillur Rahman, Shahram Bahrami, Guy Frederick Lanyon Hindley, Alexey Shadrin, Kevin S. O'Connell, Ole A. Andreassen.

**Visualization:** Toril Dammen.

**Writing – original draft:** Kristin Torgersen, Guy Frederick Lanyon Hindley, Martin Tesli, Olav B. Smeland, Toril Dammen, Ole A. Andreassen.

**Writing – review & editing:** Kristin Torgersen, Guy Frederick Lanyon Hindley, Martin Tesli, Toril Dammen, Ole A. Andreassen.

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
