## [Decision Letter · Decision Letter 0]

10 Nov 2021

Dear Dr Torgersen,

Thank you very much for submitting your Research Article entitled 'Shared genetic loci between depression and cardiometabolic traits' to PLOS Genetics.

The manuscript was fully evaluated at the editorial level and by independent peer reviewers. The reviewers appreciated the attention to an important problem, but raised some substantial concerns about the current manuscript. Based on the reviews, we will not be able to accept this version of the manuscript, but we would be willing to review a much-revised version. We cannot, of course, promise publication at that time.

If you decide to revise the manuscript for further consideration at PLOS Genetics, please aim to resubmit within the next 60 days, unless it will take extra time to address the concerns of the reviewers, in which case we would appreciate an expected resubmission date by email to plosgenetics@plos.org.

[LINK]

We are sorry that we cannot be more positive about your manuscript at this stage. Please do not hesitate to contact us if you have any concerns or questions.

Yours sincerely,

Jonathan Flint

Senior Editor

PLOS Genetics

Gregory Barsh

Editor-in-Chief

PLOS Genetics

Reviewer's Responses to Questions

**Comments to the Authors:**

Reviewer #1: General

In this paper the authors investigated overlap at the genome-wide level and in individual loci between

depression, coronary artery disease and cardiovascular risk factors using four approaches: 1) MiXeR, which identifies the number of shared genetic loci between pairs of traits; 2) conditional QQplots, which identifies pairs of traits where there is an enrichment of pleiotropic genetic effects; 3) condFDR, which increases power to identify genetic effects in each traits in a pair where there is enrichment of pleiotropic signal, and 4) conjFDR, which identifies those loci contributing to those pleiotropic signals. The authors conclude there is substantial sharing between depression and cardiovascular disease or risk factors. This is an interesting paper, but I think there needs to be some clarifications on the methods used.

Major

1. MiXeR: the authors wrote that the model fit for CAD and other CVD risk factors were not good enough to provide reliable estimates for MiXeR when testing for their shared genetic loci with depression, leaving only MiXeR results between depression and 3/10 cardiovascular phenotypes. This seems disappointing for GWAS on cardiovascular phenotypes with relatively large sample sizes (N = 200-700K). It would be great if the authors can give an explanation: what could have caused the poor model fit in all these cases? While this may already be explained in the MiXeR paper, please could the authors expand on explaining why the model didn't work specifically in their own analyses for particular traits? It could be helpful to have the model fits for each of these phenotypes in a supplementary table to demonstrate how they didn't fulfil what criteria.

2. Conditional QQ plots: the authors wrote that the conditional QQplots "show polygenic enrichment for depression with CAD and all of the CAD factors (Suppl. fig 4-21) - here do they mean all the Dep|other factor analyses or also the other|Dep analyses, or both? Suppl. fig 4-21 cover both ways, please clarify and rewrite to make this clear. Further, the authors wrote in Methods that pleiotropic enrichment "is shown by a successively leftward deflection from the null line on the conditional Q-Q plot." I don't see this in Suppl. fig 4: CAD|DEP and 12: TC|DEP, and it isn't clear to me in 8: LDL|DEP, 10: HDL|DEP; 15: DEP|TG, 16: T2D|DEP, 19: DEP|CRP, because the lines were too close together or the proportion of SNPs associated with a phenotype didn't increase as a function of the strength of the assocation with the conditional phenotype especially at more stringent p value thresholds (yellow and purple lines). How do the authors show that these successive increases in proportions of SNPs associated were significant? I think it is necessary to test for the significance of the enrichment (eg in the condFDR paper by Andreassen et al AJHG 2013) - if the authors already did this, I do not see this presented in the paper, please add explanations of how they performed this test and a supplementary table showing these results.

3. CondFDR and conjFDR: Should condFDR and conjFDR be performed only on those pairs of traits which show enrichment in pleiotropic signal through the conditional QQplots? Please clarify in paper.

4. CondFDR and conjFDR results: How many of the conjFDR loci were found in the condFDR analyses? Please show overlap and interpretations of the results, ie how many of the new loci detected using the condFDR approach (both ways for each phenotype pair) were shown by conjFDR to be shared effects (same or different directions of effect) that indicate shared mechanisms. These should be inttegrated into Table 1.

4. Replication: It would be great to see some replication of this work. The authors said the GWAS data on depression they used did not contain data from UKBiobank, in order to minimize sample overlap between DEP and other traits in their analyses - this means the authors can ask whether the condFDR/conjFDR hits replicate well in depression in UKBiobank.

Minor

1. All supplementary figures and tables need to come with some legends - without these I cannot tell what some columns of supplementary tables are referring to or how they are calculated. If these are already provided and I missed them please could the authors point out clearly where to look for them? I have not been able to find them.

2. Why were the three manhattan plots in Fig2 chosen instead of the other manhattan plots in supp figure 22-27? Is there any particular reason the authors wants to draw attention to these three conjFDR analyses?

Reviewer #2: Review is uploaded as an attachment

**Have all data underlying the figures and results presented in the manuscript been provided?**

Reviewer #1: Yes

Reviewer #2: None

PLOS authors have the option to publish the peer review history of their article (what does this mean?). If published, this will include your full peer review and any attached files.

Reviewer #1: **Yes: **Na Cai

Reviewer #2: No

---

## [Decision Letter · Decision Letter 1]

22 Mar 2022

Dear Dr Torgersen,

We are pleased to inform you that your manuscript entitled "Shared genetic loci between depression and cardiometabolic traits" has been editorially accepted for publication in PLOS Genetics. Congratulations!

Yours sincerely,

Jonathan Flint

Senior Editor

PLOS Genetics

Gregory Barsh

Editor-in-Chief

PLOS Genetics

Comments from the reviewers (if applicable):

Reviewer's Responses to Questions

**Comments to the Authors:**

Reviewer #1: I am satisfied with the authors replies to my comments and recommend acceptance.

I have a suggestion to include the author's response to my question on qqplots in the main text, as I find the response very helpful in understanding the use of qqplots for polygenic enrichments in this instance and future work. Of course the authors may choose to rephrase so that this fits their narrative well, but I think what they have written in the response is very clear.

"Q-Q plots is a hypothesis-free tool for visual assessment of enrichment and there is no well established way to quantify observed leftward deflection. The approach used to calculate significance of enrichment in the original 2013 paper (e.g. in the condFDR paper by Andreassen et al AJHG 2013) has later not been used since this approach is not fully statistically valid across all phenotypes. Later work using this method is based on visual assessment of the Q-Q plots only (1-4)."

Reviewer #2: Reviews are in the document

**Have all data underlying the figures and results presented in the manuscript been provided?**

Reviewer #1: Yes

Reviewer #2: Yes

PLOS authors have the option to publish the peer review history of their article (what does this mean?). If published, this will include your full peer review and any attached files.

Reviewer #1: **Yes: **Na Cai

Reviewer #2: No

**Data Deposition**

http://datadryad.org/submit?journalID=pgenetics&manu=PGENETICS-D-21-01304R1

**Press Queries**

---

## [Editor Report · Acceptance letter]

27 Apr 2022

PGENETICS-D-21-01304R1 

Shared genetic loci between depression and cardiometabolic traits 

Dear Dr Torgersen, 

We are pleased to inform you that your manuscript entitled "Shared genetic loci between depression and cardiometabolic traits" has been formally accepted for publication in PLOS Genetics! Your manuscript is now with our production department and you will be notified of the publication date in due course.

With kind regards,

Livia Horvath

PLOS Genetics

On behalf of:
